# Endocrine Signals Altered by Heat Stress Impact Dairy Cow Mammary Cellular Processes at Different Stages of the Dry Period

**DOI:** 10.3390/ani11020563

**Published:** 2021-02-21

**Authors:** Véronique Ouellet, João Negrao, Amy L. Skibiel, Valerie A. Lantigua, Thiago F. Fabris, Marcela G. Marrero, Bethany Dado-Senn, Jimena Laporta, Geoffrey E. Dahl

**Affiliations:** 1Department of Animal Sciences, University of Florida, Gainesville, FL 32611, USA; veronique.ouellet.6@ulaval.ca (V.O.); jnegrao@usp.br (J.N.); askibiel@uidaho.edu (A.L.S.); vlantigua@ufl.edu (V.A.L.); tfabris89@gmail.com (T.F.F.); marcela.marrero@ufl.edu (M.G.M.); bethanydado@ufl.edu (B.D.-S.); jlaporta@wisc.edu (J.L.); 2Department of Basic Sciences, Faculty of Animal Science and Food Engineering, University of Sao Paulo, Pirassununga, SP 05508-270, Brazil

**Keywords:** apoptosis, autophagy, hormones, gene expression

## Abstract

**Simple Summary:**

Late-gestation heat stress increases blood prolactin and decreases oestrogen concentrations in dry cows. These hormonal alterations may disturb mammary gland remodelling during the dry period, thereby being potentially responsible for the observed production impairments during the subsequent lactation. This project aimed to better understand the molecular mechanisms underlying subsequent impairments in mammary performance after dry period heat stress. For this, we studied the expression of genes encompassing prolactin and oestrogen pathways and key cellular process pathways under different thermal environments and in vitro hormonal milieus. The results of this study revealed that late-gestation heat stress impacted the expression of genes in the mammary gland involved in key cellular processes occurring during the dry period. Furthermore, our results indicated that these modifications are in part modulated by alterations of oestrogen and prolactin signalling.

**Abstract:**

Hormonal alterations occurring under late gestation heat stress may disturb mammary gland remodelling, resulting in a reduced milk yield during the subsequent lactation. We investigated the effects of an altered endocrine environment on mammary gene expression at different stages of the dry period. Mammary gland biopsies from in vivo-cooled (CL) or heat-stressed (HT) cows were collected at d 3 and 35 relative to dry-off and divided into explants. Explants were incubated in vitro for 24 h in one of three media: Basal: no prolactin or estrogen; CL-mimic: Basal + low prolactin + high 17β-estradiol, or HT-mimic: Basal + high prolactin + low 17β-estradiol. Real time qPCR was used to quantify gene expression. We established that late-gestation heat stress changes the expression of prolactin and oestrogen receptors, downregulates genes involved in apoptosis, autophagy and proliferation at d 3 and upregulates genes related to those cellular processes at d 35. Moreover, compared with in vivo treatments, we showed that the expression of fewer genes was impacted by in vitro treatments which aimed to mimic the hormonal response of cows exposed to a different environment. Further research will continue to uncover the mechanisms behind the production impairments caused by late-gestation heat stress.

## 1. Introduction

During the lactation cycle, the mammary gland undergoes a series of cyclical developmental changes. In dairy cows, the dry period, a non-lactating state that connects successive lactations, typically last 60 days and involve both a far-off and a close-up period that begins 3 weeks before expected calving. The dry period is important for the removal of senescent mammary cells and their regrowth before parturition [1,2]. It can be functionally divided into two phases: mammary involution, which is rapidly initiated after cessation of milk removal, and the subsequent mammary growth until calving [3]. Intracellular processes such as apoptosis [4,5] and autophagy [6] facilitate cell turnover during mammary involution, whereas mammary growth is driven by accelerated cell proliferation. Once lactation begins, the overall capacity of the mammary gland to synthesize and store milk depends on the dynamic equilibrium between cell turnover and proliferation established during the functional phases of the dry period [2].

Dry period heat stress undermines production in dairy cows [7,8]. Subsequent milk yield losses of 10% (3.6 kg/d; reviewed by Ouellet et al. [9]) are observed when cows are exposed to heat stress during the last two months of gestation. However, the underlying biological mechanisms of this decline are not yet fully elucidated. Potential explanations for the impaired mammary gland development in heat-stressed dry cows include reduced gestation length which may lead to lower milk outputs during early lactation [10,11], limited nutrient availability due to decreased dry matter intake [12,13] and redistributed blood flow to the periphery [14]. However, other factors related to altered endocrine signalling and immune function due to heat stress may also play a role in mammary cell turnover and proliferation. Circulating prolactin (PRL) concentrations increase, whereas estrone sulphate (E) concentrations decrease in cows exposed to hyperthermia during the dry period [15,16,17]. Several lines of evidence indicate that these two hormones can blunt mammary gland remodelling, as PRL serves as an anti-apoptotic factor in mammary tissue [18], whereas oestrogen promotes both autophagy and proliferation [19,20]. However, the relationship of these endocrine mechanisms to heat stress-induced shifts in mammary development is yet to be determined.

Although studies have revealed that mammary tissue from heat-stressed dry cows has lower expression of autophagy and apoptosis markers during involution [21,22,23], much remains to be understood about the effects of heat stress on mammary gland development during different stages of the dry period. Herein, we investigated the impact of different thermal environment and altered endocrine signals on expression of genes involved in PRL and E signalling, autophagy, apoptosis, and proliferation, using mammary tissue explants collected from heat-stressed and cooled dry cows during the early and late dry period. We hypothesized that heat stress-driven disturbances in hormone concentrations would alter the expression of genes involved in mammary cell autophagy, apoptosis, and proliferation during the dry period.

## 2. Materials and methods

### 2.1. Animals and Experimental Design

The experiment was conducted between May and November 2018 at the University of Florida Dairy Unit in Hague, FL, USA. All procedures were approved by the University of Florida Institutional Animal Care and Use Committee.

The experimental design for this study follows that described in Fabris et al. [23]. Briefly, twenty multiparous lactating Holstein dairy cows housed in a sand-bedded freestall barn were dried off at ~45 days prior to expected calving date based on standard operating procedures at the University of Florida Dairy Unit. Cows were randomly assigned to one of two in vivo treatments: heat stress (HT; only provided shade of the barn) or cooling (CL; provided shade of the barn and active cooling by fans and soakers), based on parity and mature equivalent milk production of the previous lactation. All cows were fed a common total mixed ration ad libitum formulated to meet the nutrient requirements of dairy cows based on Nutrient Requirements of Dairy Cattle (NRC) [24] recommendations. All cows had access to water at all times.

### 2.2. Indicators of Heat Stress

Air temperature and relative humidity of both sides of the barn were recorded every 15 min by using Hobo Pro series temp probes (Onset Computer Corp., Pocasset, MA, USA). From these variables, daily temperature-humidity index (*THI*) was calculated following the equation developed by the National Research Council (NRC) [25] and recommended for use in subtropical environments by Dikmen and Hansen [26]:THI=(1.8 x T+32)−[(0.55−0.0055 x RH)+(1.8 x T−26)]
where *T* = dry bulb temperature (°C) and *RH* = relative humidity (%).

Vaginal temperature (°C) and respiration rate (breaths/min) were also measured in all cows to confirm that cows were exposed to heat stress. Blank controlled internal drug release devices were fitted with a temperature probe (i-button cat. #DS1922-F5, accuracy ± 0.0625 °C; Maxim, Irving, TX, USA) and inserted vaginally to measure vaginal temperatures every 10 min and averaged hourly, as described by Fabris et al. [23]. Respiration rates were measured by counting flank movements for 1 min, as described by Dado-Senn et al. [27].

### 2.3. Mammary Biopsies and Tissue Explant Processing

Mammary biopsies were collected on d 3 (early mammary involution) and 35 (mammary redevelopment phase) relative to dry-off from a subset of seven cows per treatment (Figure 1). This sample size was calculated based on the mammary cell proliferation data obtained in the late dry period from heat-stressed or cooled cows [7] using a level of significance of 0.05 and 85% power. Mammary biopsies were taken from the right or left rear quarters of the udder, alternating sides for each biopsy time point following the method described by Skibiel et al. [28].

Mammary explants were prepared and cultured following procedures by Accorsi et al. [18] and by De Vries et al. [29] with modifications. Briefly, after biopsy, tissues were placed in sterilized, cold Medium 199 containing gentamicin (50 μg/mL), penicillin (100 IU/mL) and streptomycin (100 μg/mL), and immediately cut into small (<2 mm) pieces. After rinsing with sterilized Phosphate-buffered saline solution (PBS) containing amphotericin B (1.5 μg/mL), penicillin (100 IU/mL) and streptomycin (100 μg/mL), approximately 100 mg of explant were then placed in cell culture inserts (8.0 µm pore size) in a multi-well plate and cultured in a humidified chamber (95% air and 5% CO_2_) at 38.5 °C for 24 h in each of the following 3 media (in vitro treatments): (1) basal media (B): Medium 199 supplemented with 1 ng/mL bovine insulin (Sigma #I6634), and 6.5 ng/mL hydrocortisone (Sigma #H0888); (2) media (C) to mimic the hormone combination of CL cows: B + ovine pituitary PRL (20 ng/mL, Prospec-Tany Technogene, Ltd., Ness-Ziona, Israel, #cyt-240) + 17β-estradiol (5.8 ng/mL, Sigma #E2758); and (3) media (H) that mimics the hormone combination of HT cows: B + ovine pituitary PRL (40 ng/mL) + 17β-estradiol (2.9 ng/mL).

The incubation temperature was selected based on thermoneutral cow body temperature to mimic the CL environment in vivo; and an incubation time of 24 h has proven reasonable to effectively induce cellular events by addition of hormones in a mammary explant culture system [18]. The hormonal concentrations in media C or H were selected based on previously published data [8,15,16,30] to mimic the endocrine profile in the blood of CL and HT dry cows, respectively.

### 2.4. Lactate Dehydrogenase Activity Analysis

When disease or injury damages tissue, cells release LDH, which can therefore serve as index of cell injury and death. To verify that tissue was viable after the 24 h incubation, lactate dehydrogenase (LDH) activity was evaluated in: (1) media deprived of cells (B media only), (2) B media containing mammary tissue explants, and (3) B media containing 0.1% Triton-X with tissue explants. Explants with Triton-X were incubated at room temperature for 24 h and served as a positive control for maximum LDH release. Lactate dehydrogenase activity was measured in the media using a colorimetric assay kit according to the manufacturer’s instructions (BioVision, Milpitas, CA, USA). Media were harvested after the 24 h incubation and assayed immediately after. Plates were read at an absorbance at 450 nm before and after a 30 min incubation at 37 °C. The basis of this test is a coupled enzymatic reaction in which LDH present within the sample catalyses the reversible conversion of lactate into pyruvate with the concomitant reduction of NAD+ to NADH, which then interacts with a probe to produce a colour signal in proportion to LDH activity.

### 2.5. Explant RNA Extraction

Total RNA was isolated from the tissue using a Qiagen RNA extraction (RNeasy Plus Universal Mini Kit, Qiagen, cat. #73404, Venlo, The Netherlands) following the manufacturer’s instructions. The concentration and quality of extracted RNA was then determined by spectrophotometry using a NanoDrop (NanoDrop™ Spectrophotometer, Thermo Scientific, Waltham, MA, USA; #ND-2000). The concentration of total RNA ranged from 6.9 to 816.6 ng/µL, with an average (± SD) of 151.4 ± 118.9 ng/µL. The RNA was considered of sufficient purity for further analysis if the 260 nm:280 nm ratio was >1.80 (Thermo Scientific, Waltham, MA, USA). Samples were stored at −80 °C until further analysis.

### 2.6. Gene Expression Analysis

The expression of genes related to PRL and E pathways, apoptosis, autophagy, cell proliferation, inflammation, immunity, branching morphogenesis, and heat shock response was quantified in tissue explants. For this, high-throughput Multiplex RT-qPCR BioMark Dynamic Array Integrated Fluidic Circuits (IFCs) were used (Fluidigm Corporation, San Francisco, CA, USA). Briefly, 96 primers targeting 91 genes of interest, 4 reference genes, namely beta actin (*ACTB*), glyceraldehyde 3-phosphate dehydrogenase (*GADPH*), ribosomal protein S9 (*RSP9*) and hypoxanthine phosphoribosyltransferase 1 (*HPRT1*) and one structural reference gene were assayed (Appendix A). Targeted genes were selected to feature key genes in pathways that could be affected by the altered hormonal milieu caused by heat stress. Primers were designed using publicly available bovine gene sequences from the National Center for Biotechnology Information gene database (http://www.ncbi.nlm.nih.gov, accessed on 11 September 2019).

The RNA of tissue explants was diluted to 6 ng/μL. As previously described by Marrero et al. [31], all samples were normalized to 256 pg RNA and transferred to the IFC plate with the primer-probe sets. All nanolitre reactions were performed as per manufacturer’s recommendations using the following thermal protocol: 95 °C for 1 min, followed by 30 cycles of 96 °C for 5 s and 60 °C for 20 s. The software Fluidigm Real-Time PCR Analysis was used to calculate Ct values for all 96 genes for each sample. Non-detectable expression was set at a Ct of 28.92.

Relative quantification was achieved using the delta Ct (ΔCt) method [32]. The stability of the housekeeping genes was tested using geNorm. The program indicated that *ACTB*, *RSP9* and *HPRT1* were the most stable genes. Therefore, the geometric mean of the Ct values of these three most stable reference genes was calculated, and the target assay Ct values were made relative to the geometric mean of reference genes (ΔCt). Normalized gene expression (ΔCt) was used for gene expression statistical analysis.

### 2.7. Principal Component Analysis

Principal component analysis (PCA) expresses data to highlight their similarities and differences without much loss of information [33] and was herein performed to test whether the principal components (PC) would cluster according to biopsy time (i.e., mammary involution or redevelopment), gene function (where relative gene expression was averaged by function: heat shock, PRL and E signalling, cell proliferation, autophagy, apoptosis, immunity/inflammation, branching morphogenesis) or treatments (in vivo or in vitro).

### 2.8. Statistical Analyses

All statistical analyses were performed in SAS (version 9.4, SAS Institute, Inc., Cary, NC, USA). Data were first tested for covariance (Levene’s test) and normality of distribution was tested by evaluating Shapiro–Wilk statistics using the univariate procedure. Differences in THI were analysed by pen using a generalized linear mixed model with the main effects of treatment, day (repeated measure), and their interaction. Respiration rate and rectal temperature were analysed with the MIXED procedure with the main effects of treatment (CL, HT), and day (repeated) and cow ID nested within treatments as a random effect. Differences in LDH activity were analysed with PROC GLM with treatments as a fixed effect.

Normalized gene expression (ΔCt) at mammary involution and mammary redevelopment phase was evaluated separately. The model included in vivo treatment (CL or HT), in vitro treatments (B, C, H) and their interaction as fixed effects. *p* values ≤ 0.05 were considered statistically significant and *p*-values ≤ 0.15 were considered a trend toward significance. The estimates of the model for each gene were then used to calculate fold change (ΔΔCt). To evaluate the effects of the in vivo treatments (CL, HT) on gene expression, the fold change relative to the CL cows (ΔΔCt) was calculated as shown in Equation [1]:ΔΔCt (in vivo) = 2 − (ΔCt HT − ΔCt CL).(1)

To evaluate the effects of the in vitro treatments (B, C and H) on gene expression, the fold change relative to mRNA abundance quantified in explants cultures in media B was calculated as shown in Equations (2) and (3):ΔΔCt (in vitro) = 2 − (ΔCt C − ΔCt B). (2)
ΔΔCt (in vitro) = 2 − (ΔCt H − ΔCt B). (3)

## 3. Results

### 3.1. Thermal Environment and Physiological Parameters In Vivo

Throughout the dry period, THI measurements recorded at the experimental pens level were similar between CL and HT treatments (78.21 vs. 78.76 ± 0.25, *p* = 0.24). However, relative to HT, CL cows had X °C lower vaginal temperatures (38.9 vs. 39.2 ± 0.2 °C, *p* < 0.01). Additionally, CL cows had lower respiration rates compared to their HT counterparts (54.6 vs. 65.3 ± 1.2 breaths/min, *p* < 0.05).

### 3.2. Lactate Dehydrogenase Activity In Vitro

Lactate dehydrogenase activity (mmol/min/mL) was lower in B media deprived of tissue explants relative to when mammary tissue explants were incubated in B media (0.11 vs. 2.54 ± 0.4 mmol/min/mL; *p* < 0.05; Figure 2). Higher LDH activity (mmol/min/mL) was observed in cells incubated in B media with 0.1% Triton-X (Thermo Fisher Scientific Inc., Waltham, MA, USA), which served as the maximum release sample, compared with tissue explants incubated in the B media (6.06 vs. 2.54 ± 0.7 mmol/min/mL; *p* < 0.05; Figure 2).

### 3.3. Principal Component Analysis

The PCA demonstrated that the expression of mammary tissue harvested during early mammary involution and mammary redevelopment clustered distinctively (Figure 3a). The first two PC explained 60% of total variation (40% and 20% for PC1 and PC2, respectively) when the statistical analysis of differences in gene expression was performed by biopsy time point separately. Furthermore, when the relative expression of genes was averaged by functional pathway, the PCA loading plot showed that the relative gene expression clustered by gene function, where the first two PCs explained 80% of total variation (53% and 27% for PC1 and PC2, respectively) (Figure 3b). Finally, when averaged by treatments, the PCA indicated that relative expression clustered by in vivo and in vitro treatments with the first two PC explained 58% of total variation (38% and 20% for PC1 and PC2, respectively) (Figure 3c).

### 3.4. Effects of In Vivo Thermal Environment on Mammary Gland Gene Expression

Heat exposure during the dry period impacted the expression of prolactin receptor long form (*PRLR long*; *p* = 0.02), serine/threonine kinase B-raf (*BRAF*; *p* = 0.09) and oestrogen receptor 2 (*ESR2*; *p* = 0.04) as those genes involved in PRL and E signalling tended to be or was upregulated in HT explants compared to CL explants during mammary involution (Figure 4a,c). During mammary redevelopment, *PRLR long* was upregulated, whereas RAS p21 protein activator 1 (*RASA1*), *BRAF*, and *ESR2* tended to be or were downregulated in HT explants relative to CL explants (Figure 4a,c, all *p* ≤ 0.10).

Heat stress exposure during the dry period altered the expression of 9 out of the 13 apoptosis-related genes evaluated (Figure 4b). For example, in vivo heat stress exposure downregulated or tended to downregulate the expression of Apoptotic peptidase activating factor 1 (*APAF1*; *p* = 0.15), Caspase 3 (*CASP3*; *p* = 0.07), Caspase 9 (*CASP9*; *p* = 0.05) during mammary involution, whereas insulin-like growth factor binding protein 3 (*IGFPB3*; *p* = 0.05) was upregulated. During the mammary redevelopment phase, the gene expression of apoptosis inducing factor mitochondria associated 1 (*AIFM1*; *p* = 0.06), *APAF1* (*p* = 0.05), apoptosis regulator *BAX* (BAX; *p* = 0.01), *IGFBP3* (*p* = 0.05), and insulin-like growth factor binding protein 5 (*IGFBP5*; *p* = 0.12) tended to be or were upregulated, whereas the expression of prostaglandin E synthase (*PTGES*; *p* = 0.14) tended to be downregulated in HT explants relative to CL explants.

The thermal environment to which the animals were subjected during the dry period also impacted the gene expression of 4 out of 11 autophagy-related genes analysed (Figure 4d). During mammary involution, Autophagy related AGT5 (*ATG5*; *p* = 0.15) and Beclin 1 (*BECN1*) tended (*p* = 0.07) to be downregulated, whereas microtubule-associated protein 1 light chain 3 beta (*MAP1LC3A*) was upregulated (*p* = 0.02) in HT explants relative to CL explants. Furthermore, during the mammary redevelopment phase, *MAP1LC3A* was upregulated (*p* = 0.01) in HT explants.

Few immune related genes in the mammary tissue were impacted by the thermal environment. Specifically, nuclear factor-kappa B (*NF-κB*; *p* = 0.05) was, and transforming growth factor beta 3 (*TGFB3*; *p* = 0.07) tended to be upregulated in HT explants during mammary involution, whereas CD4 molecule (*CD4*; *p* = 0.12) tended to be upregulated during mammary redevelopment phase. Moreover, heat shock response related gene heat shock protein family A HSP70 (*HSPA1A*; *p* = 0.02) and ductal branching-related gene Tektin 3 (*TEKT3*; *p* = 0.07) were respectively upregulated or tended to be upregulated in HT explants during mammary redevelopment (Figure 4f).

Heat stress exposure during the dry period also impacted the expression of 9 out of 22 genes involved in cell proliferation evaluated herein (Figure 4g). More specifically, AKT serine/threonine kinase 2 (*AKT2*; *p* = 0.08) tended to be upregulated during early mammary involution, whereas epidermal growth factor (*EGF*; *p* = 0.10), mitogen-activated protein kinase 3 (*MAPK3*; *p* = 0.04), pyruvate dehydrogenase 1 (*PDK1*; *p* = 0.10), phosphoinositide-3-kinase regulatory subunit 2 (*PIK3R2*; *p* = 0.10), and Ras homolog mTOR1 binding (*RHEB*; *p* = 0.15) tended to be or were downregulated in HT explants. In addition, Cyclin D1 (*CCND1*; *p* = 0.06) and TEA domain transcription factor 4 (TEAD4; *p* = 0.15) tended to be upregulated in HT explants during the mammary redevelopment phase of the dry period.

### 3.5. Effects of In Vitro Treatments on Mammary Gland Gene Expression

The expression of several genes involved in PRL and E signalling was impacted by our in vitro treatments during mammary involution, whereas no genes involved in PRL and E signalling were impacted during the mammary redevelopment phase. More specifically, casein beta (*CSN2*; *p* = 0.10), HRAS proto-oncogene, GTPase (*HRAS*; *p* = 0.08), Raf-1 proto-oncogene, serine/threonine kinase (*RAF1*; *p* = 0.04), suppressor of cytokine signaling 2 (*SOCS2*; *p* = 0.05), Signal Transducers and Activators of Transcription 1 (*STAT1*; *p* = 0.12) and Signal Transducers and Activators of Transcription 3 (*STAT3*; *p* = 0.04) were or tended to be downregulated in explants cultured in C media relative to when explants are cultured in B media. All of these aforementioned genes were also downregulated in explants cultured in H media relative to when explants were cultured in B media. However, only *SOCS2* (*p* = 0.05) and *STAT3* (*p* = 0.05) were significantly downregulated. In addition, *BRAF* (*p* = 0.01), oestrogen receptor 1 (*ESR1*; *p* = 0.10), *ESR2* (*p* = 0.02), G protein-coupled oestrogen receptor (*GPER1*; *p* = 0.12), Mitogen-activated protein kinase 1 (*MAP2K1*; *p* = 0.06), and mitogen-activated protein kinase 1 (*MAPK1*; *p* = 0.04) (all involved in E signalling) genes were or tended to be downregulated in explants cultured in C media relative to explants cultured in B media. Of these genes, *BRAF* (*p* = 0.01), *MAP2K1* (*p* = 0.06), and *MAPK1* were also downregulated, whereas *ESR2* was upregulated in explants cultured in H media compared to explants cultured in B media (Figure 5c).

Out of the 13 apoptosis-related genes (Figure 5b) and of the 11 autophagy-related genes (Figure 5d) analysed, only the mRNA abundance of BCL2-associated agonist of cell death (*BAD*), Fas-Ligand (*FASLG)*, autophagy-related ATG3 (*ATG3)*, autophagy-related ATG7 (*ATG7)*, microtubule-associated protein 1 light chain 3 beta *(MAP1LC3B)*, and mitogen-activated protein kinase 14 (*MAPK14*) were impacted by the in vitro treatments. More specifically, during mammary involution, autophagy-related genes autophagy-related ATG3 (*p =* 0.03) and autophagy-related ATG7 (*p =* 0.07) were or tended to be upregulated in explants cultured in C media relative to explants cultured in B media. During the mammary redevelopment phase of the dry period, *BAD* (*p* = 0.14), *FASLG* (*p* = 0.02), and *MAP1LB3B* (*p* = 0.04) were or tended to be upregulated in explants cultured in H media relative to explants cultured in B media, whereas *MAPK14* (*p* = 0.02) was downregulated. In addition, *FASLG* tended to be upregulated in explants cultured in C media relative to explants cultured in B media.

Out of the 22 genes involved in cell proliferation that were measured, only two were impacted by in vitro treatment (Figure 5f). During mammary involution, chemokine ligand 2 (*CXCL2*) tended to be downregulated (*p* = 0.14) in explants cultured in C media relative to explants cultured in B media. During the mammary redevelopment phase of the dry period, phosphoinositide-3-kinase regulatory subunit 2 (*PIK3R2*; *p* = 0.02) was upregulated in explants cultured in H media relative to explants cultured in B media.

Furthermore, 5 out of the 11 immune-related genes were impacted by the in vitro treatments (Figure 5e). During mammary involution, interleukin 1 alpha (*IL1A*; *p* = 0.08) and interleukin 1 beta (*IL1B*; *p* = 0.01) tended to be or were downregulated in explants cultured in C and H media relative to explants cultured in B media. During mammary redevelopment, C-X-C motif chemokine ligand 8 (*CXCL8*; *p* = 0.07), *IL1A* (*p* < 0.001), *IL1B* (*p* = 0.06), interleukin 6 (*IL6*; *p* = 0.01), and tumor necrosis factor alpha (*TNF*; *p* = 0.01) were or tended to be downregulated in explants cultured in C and H media relative to explants cultured in B media.

### 3.6. Interactions between In Vivo and In Vitro Treatments

A significant interaction between in vivo and in vitro treatments was identified for 8 out of 91 targeted genes (Table 1). For example, during mammary involution, *BRAF* and *ESR2,* which are involved in E signalling, displayed a significant (both *p* = 0.03) interaction between the in vivo and in vitro treatments. During the redevelopment phase of the dry period, hydroxysteroid 17-beta dehydrogenase 6 (*HSD17B6*; E signalling), caspase 8 (*CASP8*; apoptosis), *MAP1LC3B* (autophagy), lipopolysaccharide binding protein (*LBP*; inflammation), vascular endothelial growth factor (*VEGFA*; morphology), and *PIK3R2* (cell proliferation) displayed a significant interaction between the in vivo and in vitro treatments.

## 4. Discussion

In the mammary gland, pro-apoptotic and autophagic signalling factors are triggered after the cessation of milk removal during the early phase of the dry period (involution), whereas the mammary redevelopment phase is characterized by upregulation of proliferation related genes [34]. Any disturbances in these cellular processes may severely impede the mammary gland’s ability to synthesize milk in the subsequent lactation. During the past ten years, it has been well documented that dry period exposure of dairy cows to heat stress negatively impacts the next lactation (reviewed by Ouellet et al. [9]). Herein, we aimed to gain a better understanding of the underlying mechanisms for the observed impairments after dry period heat stress, by studying the mammary gene expression related to PRL and E pathways and key cellular process pathways under different thermal environments and hormonal milieus.

Temperature-humidity index measurements indicated that both CL and HT cows were under similar thermal environmental conditions during late gestation. The increase in vaginal temperature and respiratory frequency suggested that HT cows were not effective in maintaining euthermia and that the cooling system of fans and soakers was effective in alleviating heat stress in CL cows. Thus, in vivo treatments were properly induced in the current study. The quantification of LDH activity confirmed that the in vitro protocol (i.e., incubation time, temperature and hormonal concentrations) applied to mammary tissue explants was appropriate to maintain tissue viability. Therefore, our mammary tissue explants were a suitable experimental model to study hormonal mechanisms induced by heat stress.

Principal component analysis is often used to transform original dependent variables into new uncorrelated dimensions to simplify data structure, eliminate descriptor redundancies, and indicate potential latent causal variables [35]. In the current study, PCA was applied to the normalized relative gene expression to simplify the interpretation of data. As the first two PC accounted for explained 60% of total variation when the statistical analysis of differences in gene expression was performed by biopsy time point separately, further analysis was conducted by biopsy time separately. Therefore, the PCA gave insight on how to structure the data analysis given the complexity of the experimental design.

In the present study, the in vivo-imposed heat stress significantly influenced the expression of *PRLR long* and *ESR2* genes in mammary tissue explants during mammary involution. Previous studies have reported that heat stress induced during the dry period decreases E and increases PRL in plasma when compared to cows in thermoneutral or cool conditions [15,16,17]. The exact reason why heat stress modulates PRL and E remains unknown. Research suggests that the increase in PRL under heat stress may be associated with the sweating response, the induction of heat shock proteins and pelage molting, all mechanisms expected to improve heat loss capacity [36]. Our results show that heat stress alters the expression of PRL and E receptors in the mammary gland relative to cool conditions, suggesting that heat stress during the dry period changes the responsiveness of the mammary cells to E and PRL. Notably, the most pronounced PRL-induced gene in mammary tissue, casein beta (*CSN2*) [37], was not differentially expressed due to in vivo treatments in the current study. This result is not in accordance with Fabris et al. [23] who reported that *CSN2* was upregulated in cows exposed to HT during the late dry period. The discrepancy may be attributed to the differed timing of mammary biopsies. However, our findings are in accordance with Corazzin et al. [38], who reported similar levels of *CSN2* in milk somatic cells (representative of mammary tissue) of lactating cows exposed to thermoneutral conditions and then mild heat stress.

Emerging evidence indicates that PRL and E modifications have the potential to alter key cellular processes, such as apoptosis and autophagy occurring during the mammary dry period. Apoptosis and autophagy are important mechanisms for mammary epithelial cell death, and consequently, for cell renewal throughout the dry period [6,20]. Prolactin was shown to decrease mammary cell apoptosis in vitro by optimizing the action of IGF-1, a growth factor, through downregulation of *IGFBP5* gene expression [6] and E positively regulates autophagy in vitro in bovine mammary epithelial cells, likely via membrane-bound receptors [39]. In the current study, in vivo heat stress significantly impacted apoptosis and autophagy pathways during mammary involution, which is consistent with previous studies from our laboratory [7,21,23]. However, heat stress had no significant effect on *IGFBP5* expression during mammary involution, but it tended to be upregulated in HT explants during the mammary redevelopment phase of the dry period, which could be associated with disturbed cell proliferation as calving approaches. Furthermore, this could potentially indicate that PRL may modulate apoptosis in vivo in more than one way. Herein, no heat shock protein (HSP) genes were significantly impacted by the imposed in vivo treatments. However, this was expected as HSP related genes are associated with acute heat stress, whereas the animals of the current study were exposed to chronic heat stress [40]. Heat shock proteins act as molecular chaperones to assist protein folding under heat stress conditions and interfere with major apoptotic pathways to protect cells from hyperthermia [40,41]. Moreover, recent investigations have also demonstrated an inverse relationship between the induction of HSP70 and autophagy induction in human cells [42]. Collectively, in the current study, most genes related to apoptosis and autophagy that were significantly impacted by in vivo heat stress were downregulated in HT explants relative to CL explants during early involution. However, those genes were upregulated during mammary redevelopment which lends credence to the hypothesis that heat stress may alter the trajectory of apoptosis and autophagy in mammary tissue explants during early involution and even during the late dry period.

In contrast to the involution period, the mammary redevelopment phase is characterised by rapid cell proliferation. In the present study, most proliferation-related genes analysed were not different between in vivo treatments. However, the few that were differentially expressed were downregulated in HT explants relative to CL explants during mammary involution, then surprisingly upregulated during mammary redevelopment (i.e., *CCDN1*, *TEAD4*). Therefore, our results could potentially indicate that apoptosis and autophagy are more impacted by heat treatments than cell proliferation, which is in line with recent findings of Fabris et al. [23].

Cessation of milking also induces the recruitment of immune cells and immune factors that in turn offer protection against inflammation [3]. In the present study, genes related to immunity and inflammation were upregulated in HT explants relative to CL explants during both mammary involution and redevelopment, indicative of compromised immunity and inflammation in HT cows throughout the dry period. Impaired immunity under heat stress was previously reported by Thompson and Dahl [43], who examined the impact of dry period season as a proxy for heat stress effects. Those authors reported that high ambient temperatures reduced the immune competence of cows, as dry periods corresponding to the summer months increased the incidence of mastitis, respiratory disease, and displaced abomasum. Furthermore, a porcine study demonstrates plausible triggering the inflammatory response under heat stress [44] whereby *TNF* intramuscular relative abundance was increased following 24 h of heat stress, possibly due to migration from the vasculature into tissues. In parallel, increased oxidative stress was also detected. Increased *TNF* expression or oxidative stress would ostensibly initiate an inflammatory response via *NF-κB*, which could also be the case in the current study, as NFK relative abundance was increased in HT explants.

Irrespective of the in vivo treatments, in vitro treatment that mimics the hormonal responses of heat-stressed or cooled cows also significantly affected the expression of several genes encompassing PRL and E signalling, apoptosis, autophagy, proliferation, immunity and inflammation pathways. Interestingly, the most significant effects of in vitro treatments were reported during mammary involution. Furthermore, relative to explants cultured in B media, which did not contain PRL and E, the majority of genes studied were downregulated in explants cultured in C and H media. In fact, our results showed that the two in vitro treatments consisting of two different hormonal combinations similarly impacted the expression of *STAT3*, *MAPK1*, and *MAP2K1* (that regulate PRL and E signalling and cell survival pathway), *SOCS2* and *BRAF* genes (that regulate PRL signalling and apoptosis pathway), and *IL1A*, *IL1B* (immunity pathway) during mammary involution. Furthermore, expression of *CSN2*, *HRAS*, *RAF1*, *STAT1* (all involved in PRL signalling), *ESR1*, *ESR2*, *GPER1* (involved in E signalling), *ATG3*, *ATG7* (both involved in autophagy) and *CXCL2* (cell survival) were impacted differently by the different hormonal combinations tested in vitro. Since mammary involution is hallmarked by intense cellular turnover due to increases in apoptosis and autophagy [45] and immune cell recruitment [3], we speculate that the interaction between these changes in gene expression has different impacts on the apoptotic, autophagic, and proliferative rates of the H and C treatments thereby resulting in different lactational performance in the subsequent lactation. It is noteworthy to mention that no apoptosis-related gene was impacted by the tested in vitro treatment during mammary involution. This could potentially indicate that other mechanisms besides PRL and E alterations may modulate apoptosis in the early dry period.

Relative to mammary involution, fewer genes were impacted by the in vitro treatments tested during the mammary redevelopment phase of the dry period. Interestingly, the expression of genes related to immunity and inflammation was downregulated irrespective of the culture media (C or H), indicating that PRL and E hormonal alterations may impact these pathways in a permissive manner. Prepartum heat exposure was previously reported to depress immune function and evidence links this decrease to altered PRL signalling under heat stress [46]. Our results lend credence to this hypothesis. Moreover, H and C in vitro treatments differently affected gene expression of *BAD* (apoptosis pathway) and *MAPK14* (autophagy pathway), which potentially indicates that change in relative levels of mammogenic hormones under heat stress may disturb autophagy and apoptosis as calving approaches, which may have repercussions during lactation.

## 5. Conclusions

Our results confirmed that exposure to heat stress during the last six weeks of gestation impacts mammary gland expression of genes involved in PRL and E signalling, apoptosis, autophagy, proliferation, immunity and inflammation. Furthermore, we demonstrated that our in vitro treatments that mimic the hormonal responses of heat-stressed or thermoneutral cows also significantly affected the expression of several genes encompassing PRL and E signalling, apoptosis, autophagy, proliferation, immunity and inflammation pathways. The fact that several genes encompassing key cellular processes were impacted differently by the two in vitro treatments may indicate that PRL and E alterations can modulate the production impairments caused by late-gestation heat stress. However, that several genes were impacted in a similar manner or not impacted by the in vitro treatments suggests that there are likely additional contributing factors under dry period heat stress beyond the direct effects of PRL and E on mammary cell death and proliferation.

## Figures and Tables

**Figure 1 animals-11-00563-f001:**
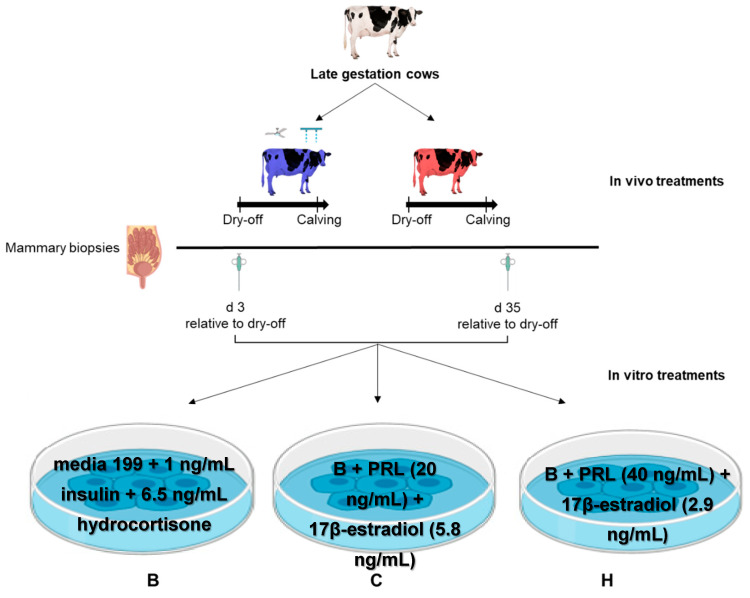
Schematic representation of the experimental design presented in the current study. Late gestation cows were dried off at 46 d before calving and assigned to 1 of 2 in vivo treatments: cooling (CL) (*n* = 7); provided shade of the barn and active cooling by soakers and fans (blue); heat treatment (HT) (*n* = 7); only provided shade of the barn (red). Mammary biopsies were collected at 3 d (to capture the involution phase) and 35 (to capture the mammary redevelopment phase) and tissue explants were incubated in 1 of 3 in vitro treatments: (1) Basal (**B**): no prolactin or oestrogen; (2) CL-mimic (**C**): B + 20 ng/mL prolactin + 5.8 ng/mL oestrogen or (3) HT-mimic (**H**): B + 40 ng/mL prolactin + 2.9 ng/mL oestrogen.

**Figure 2 animals-11-00563-f002:**
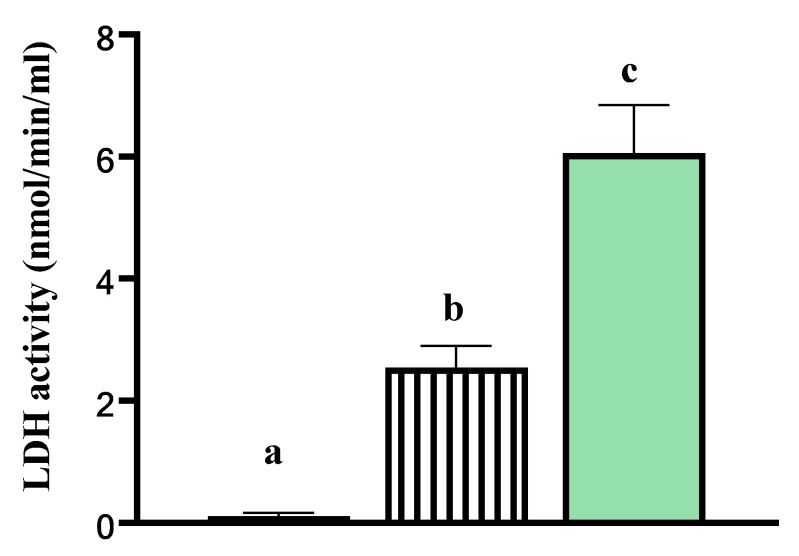
Lactate dehydrogenase (LDH) activity (mmol/min/mL) from basal media alone (
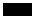
), tissue explants cultured in media (
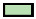
), and tissue explants incubated in Triton X-100 (
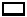
). Data are presented as least square means ± standard error of the mean (SEM) Different letters a–c indicate *p*-value < 0.05. LDH activity was higher in the media of explants incubated in Triton X-100 at room temperature relative to both the media from tissue explants cultured in vitro and relative to the basal media deprived of tissue explants. This indicated that cells within the explants cultured in vitro for 24 h were viable.

**Figure 3 animals-11-00563-f003:**
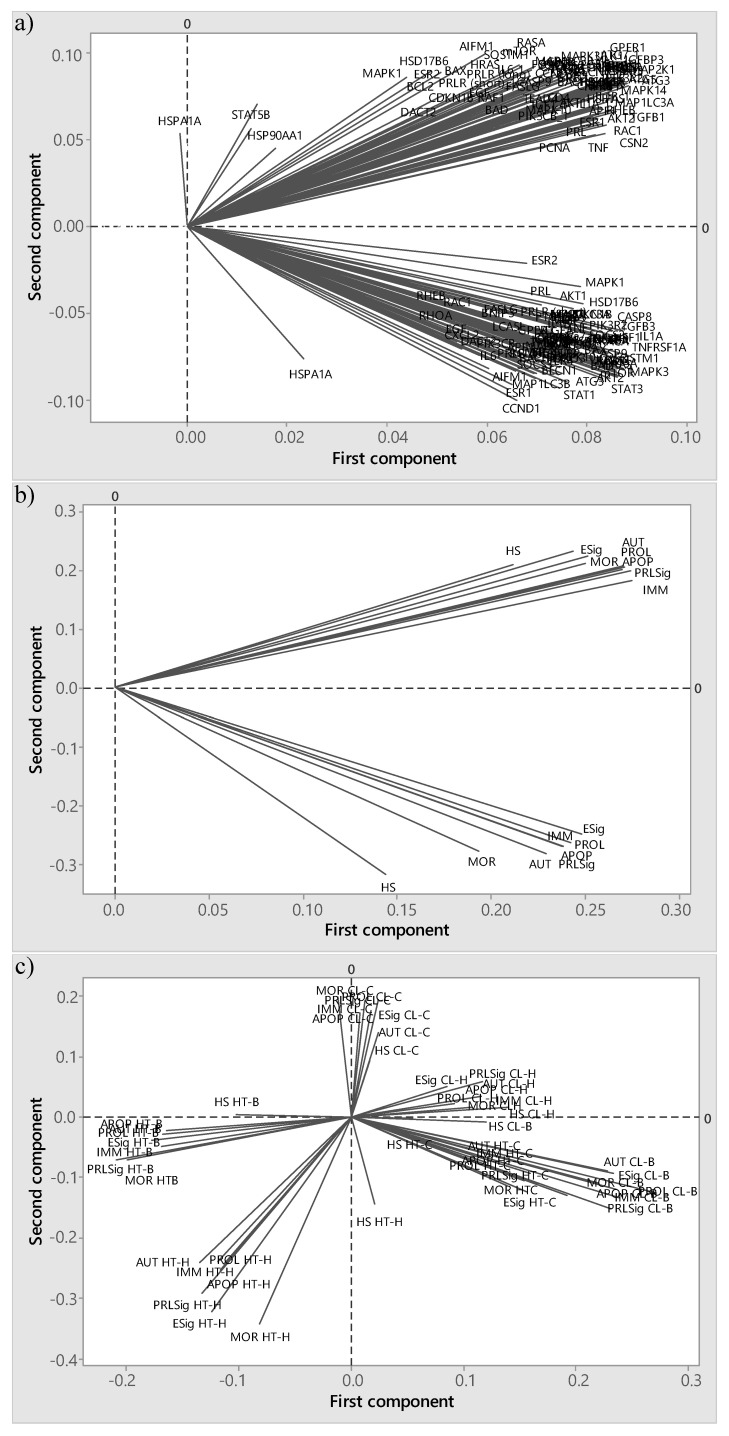
Loading plots describing relationships between: (**a**) mammary gland relative expression of 91 genes (as detailed in Appendix A) and biopsy collection time point, (**b**) gene functions: heat shock (HS), prolactin signalling (PRLSig), oestrogen signalling (ESig), cell proliferation (PROL), autophagy (AUT), apoptosis (APOP), morphology (MOR), and immunity (IMM), and (**c**) in vivo and in vitro treatments from a principal component analysis. Figure 3c included genes averaged heat shock (HS), prolactin signalling (PRLSig), oestrogen signalling (ESig), cell proliferation (PROL), autophagy (AUT), apoptosis (APOP), morphology (MOR), and immunity (IMM), by in vivo and in vitro treatments CL-H, CL-C, CL-B, HT-H, HT-C, and HT-B.

**Figure 4 animals-11-00563-f004:**
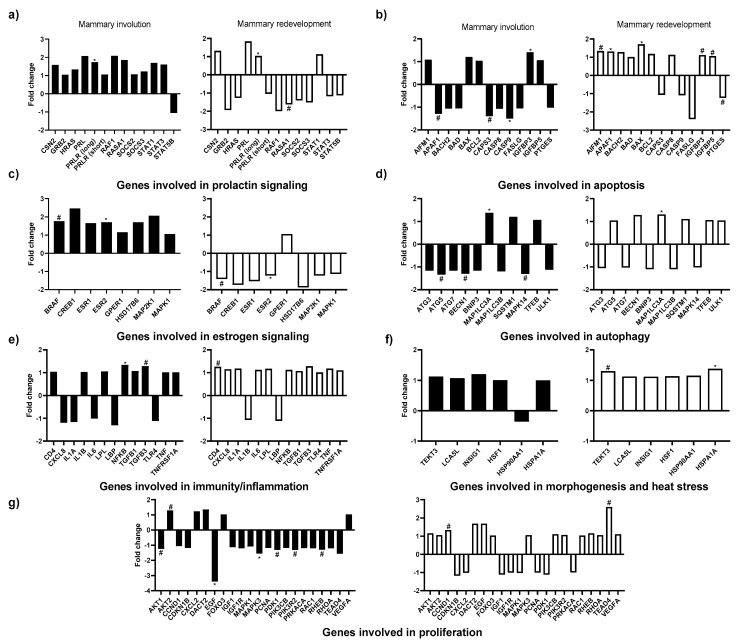
Gene expression in mammary gland tissue explants of heat-stressed dairy cows (HT) or cows provided active cooling (CL) during the dry period. Explants were collected during early involution (black columns, 3 d after dry off) and during mammary growth (white columns, 35 d after dry off) phases. Gene expression is reported as fold change (2^-ΔΔCt^) relative to cooled cows. An asterisk (*) indicates significant differences (*p* ≤ 0.05) and (#) indicates tendencies (0.05 < *p* ≤ 0.15) between HT and CL treatments. (**a**–**g**) refer to the gene function of interest.

**Figure 5 animals-11-00563-f005:**
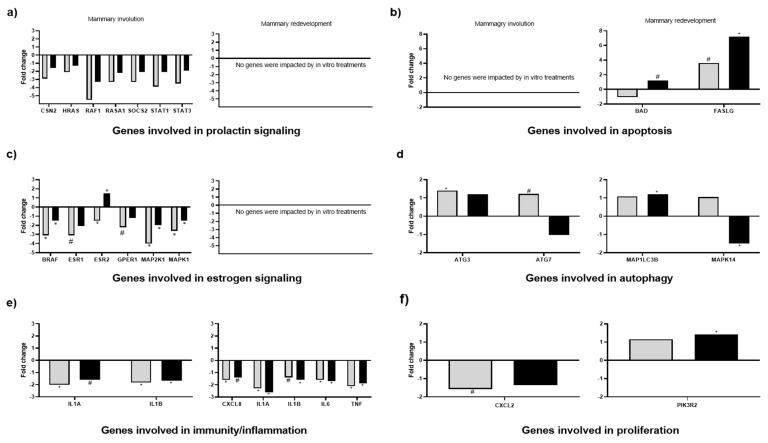
Expression of genes involved in Prolactin (PRL) and oestrogen (E) signalling, cell apoptosis in tissue explants of mammary gland incubated in 3 different media: (1) Basal (B): no PRL or E; (2) Cooled-mimic (C): B + 20 ng/mL PRL + 5.8 ng/mL E or (3) Heat stress-mimic (H): B + 40 ng/mL PRL + 2.9 ng/mL E. Explants were collected during early involution (3 d after dry off) and during mammary growth phases (35 d after dry off). Gene expression in mammary gland tissue explants is reported as fold change (2^-ΔΔCt^) relative to B media. Grey bars indicate C fold change and black bars indicate H fold change. An asterisk (*) indicates significant differences (*p* ≤ 0.05) and (#) indicates tendencies (0.05 < *p* ≤ 0.15) between C (light grey bars) and B and HT (solid black bars) and B media. (**a**–**f**) refer to gene function.

**Table 1 animals-11-00563-t001:** Relative gene expression of mammary gland tissue explants collected during involution and mammary redevelopment of the dry period in cows that were either cooled (shade, fans and soakers) or heat-stressed (only shade). Tissue explants were incubated in vitro in media with different hormone concentrations: Basal media: no prolactin (PRL), no estrogen (E); media C that mimics the hormone combination of cooled cows: Basal + low PRL + high E; media H that mimics hormone combination of heat-stressed cows: Basal + high PRL + low E. Relative gene expression is expressed as a fold change compared with Basal media.

		In Vivo-Cooled Cows		In Vivo Heat-Stressed Cows		
	Genes ^1^	C-Fold Change	H-Fold Change	C-Fold Change	H-Fold Change	*p* Value
Early involution						
E signalling						
	*BRAF*	−1.13	−1.10	−9.32	−2.39	0.03
	*ESR2*	1.03	1.04	−2.59	1.89	0.03
Mammary growth						
E signalling						
	*HSD17B6*	1.80	−2.33	1.95	14.93	0.14
Apoptosis						
	*CASP8*	−1.38	−1.06	1.36	1.24	0.06
Autophagy						
	*MAP1LC3B*	1.12	1.82	1.13	1.03	0.08
Inflammation						
	*LBP*	−1.09	1.33	1.37	−1.05	0.01
Morphology						
	*VEGFA*	1.31	1.06	−1.35	1.34	0.08
Proliferation						
	PIK3R2	1.12	1.82	1.18	1.11	0.05

^1^ Only includes target genes with significant (*p* < 0.05) interactions or tendencies (0.05 < *p* < 0.15) towards an interaction between in vivo and in vitro treatments.

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
