# Peer review of "Endocrine Signals Altered by Heat Stress Impact Dairy Cow Mammary Cellular Processes at Different Stages of the Dry Period"

_animals, 2021, doi:10.3390/ani11020563_

Round 1

Reviewer 1 Report

The study aimed to investigate the expression of genes encompassing prolactin and estrogen pathways and key cellular process pathways under different thermal environments and in vitro hormonal milieus. The results found that late-gestation heat stress impacted the expression of genes in the mammary gland involved in key cellular processes occurring during the dry period. This is a interesting study. However, the quality of the figure 3 is not good, I suggest to revise it.

Author Response

Au: Thank you for taking the time to review our work. As suggested by the reviewer, we improved to quality of Figure 3. Components lines are now thinner to enhance the clarity of the figure 3 (please see page 7).

Reviewer 2 Report

Manuscript title:

Endocrine signals altered by heat stress impact dairy cow mammary cellular processes at different stages of the dry period

The above manuscript came to me for reviewing. Authors provided insights regarding the changes in endocrine due to heat stress triggered by related mammary gene expression to investigate their impact on subsequent productivity of the dry cows.  It is of current hot topic in scientific community. This is by far one of the best research report that I have read since two years ago with respect to the novelty, originality and importance of the research. The content is very well describing the current hot topic in the area. This manuscript is very well written in case of all sections and the conclusion support the hypothesis made in the Introduction and the content of the manuscript.

My only important concern is the Abstract content. I understand the limitation of only 200 words in the Abstract but authors should try harder to cover the important results in the Abstract to reflect the whole paper contents (If possible please add/remove some of the sentences). Other than this, I have no major issues and I recommend the highest priority possible for publication of this paper after minor revision. Meanwhile, I will ask Editors to extend the Abstract limitations to 250 words if possible and if not, to make some exceptions for such good findings.

I am happy with the flow of the Introduction and the Discussion. Some figures should be modified in order to be clearer in white and black printing pages that I will address below. Some formatting issues are identified and I will address them in detailed comment. Please follow the guideline of the journal for paper preparation more strictly.

 I have only few minor corrections to ask from the authors. Detail comments are as follow:

Minor comments

Please note that all subheading word should start with capital alphabets based on the journal style. Please do this for the entire manuscript.

Abstract:

Authors used the abbreviation of (HT) for “heat stress” while they continued to use the full-term throughout the manuscript including line 31, 37, 52, 54, 56, 60 and many others. Please use the abbreviation that you defined first for the entire manuscript.

Please define 6 weeks of dry-off by providing explanation of far-off (~40 d of dry period) and close-up (~20 d of dry period), by providing a reference if applicable. It can be defined in the Introduction, or M&M section either section is okay. Only one line is enough. It is because of its importance for non-expert and expert readers to know the effective period in this study was including both far-off (in part) and close-up (in whole).

Introduction:

I stated above to be considered if applicable. Also please go ahead and use the abbreviation term of HT as defined in the Abstract.

Materials and methods:

Provided enough details and no additional input is required except modifications in Figure 1.

Figure 1. is not clear in black and white printing form. Please make sure to justify the “mammary biopsies image and also the font color of the in vitro plates (B, C, and H) to be clearer in black and white print.

Results:

The content is clear and no changes are required.

Figure 2: the Y column’s digits are not clear. Please justify

Figure 3: I hope there is a way to clarify a) and c) graph to be more readable. But I understand if it is not possible due to lots if info input. Just in case if possible, please do it so. If not, leave it as it is.

Discussion:

Very well discussed the obtained results. I enjoyed reading this section.

Conclusion:

It supports the hypothesis and the results of this study.

References:

The formatting of the References is not provided based on the journal guideline and style. Please re-do.

Example:

Tao, S.; Monteiro, A.P.A.; Thompson, I.M.; Hayen, M.J.; Dahl, G.E. Effect of late-gestation maternal heat stress on growth and immune function of dairy calves. Journal of Dairy Science 2012, 95, 7128-7136, doi:https://doi.org/10.3168/jds.2012-5697.

Author Response

Au: Thank you for taking the time to review our work and for your suggestions. The number of words included in the abstract is 200. Unfortunately, we were not able to include more words and explain more in depth our findings. However, we think that the information presented is sufficient for the readers to understand our work.

R2: Please note that all subheading word should start with capital alphabets based on the journal style. Please do this for the entire manuscript

Au: All subheadings were modified according the journal’s guidelines.

R2: Authors used the abbreviation of (HT) for “heat stress” while they continued to use the full-term throughout the manuscript including line 31, 37, 52, 54, 56, 60 and many others. Please use the abbreviation that you defined first for the entire manuscript.

Au: Thank you for the comment. We did use the abbreviation HT to refer to heat-stressed cows which is one of our in vivo treatment. However, when we refer the general terms of ‘’heat stress’’ we chose not to use the abbreviation to avoid confusion.

R2: Please define 6 weeks of dry-off by providing explanation of far-off (~40 d of dry period) and close-up (~20 d of dry period), by providing a reference if applicable. It can be defined in the Introduction, or M&M section either section is okay. Only one line is enough. It is because of its importance for non-expert and expert readers to know the effective period in this study was including both far-off (in part) and close-up (in whole).

Au:Suggestion was followed accordingly. Please see new lines 43-44.

R2: I stated above to be considered if applicable. Also please go ahead and use the abbreviation term of HT as defined in the Abstract

Au: Thank you for the comment. We did use the abbreviation HT to refer to heat-stressed cows which is one of our in vivo treatment. However, when we refer the general terms of ‘’heat stress’’ we chose not to use the abbreviation to avoid confusion.

R2: Provided enough details and no additional input is required except modifications in Figure 1.

Au:Figure 1 was changed according to suggestions. Please see lines 113-127.

R2 : Figure 2: the Y column’s digits are not clear. Please justify

Au:Figure 2 was changed according to suggestions. (please see page 6)

R2 : Figure 3: I hope there is a way to clarify a) and c) graph to be more readable. But I understand if it is not possible due to lots if info input. Just in case if possible, please do it so. If not, leave it as it is.

Au: Thank you for taking the time to review our work. As suggested by the reviewer, we improved to quality of Figure 3. Components lines are now thinner to enhance the clarity of the figure 3 (please see page 7).

R2: The formatting of the References is not provided based on the journal guideline and style. Please re-do.

Au: References format was re-done according to the journal’s guidelines.

Reviewer 3 Report

Please provide further information on the frequency of vaginal temperature (°C) and respiration rate (breaths/min) and also describe the protocol of water access for the cows. Additionally, please include measurements of mean individual heat stress, if available. Are there any data regarding heat stress conditions that the cows were exposed to before the onset of the experiment? Please explain in greater detail the environmental conditions and the fluctuation of THI within 24h.

Author Response

R3 : Please provide further information on the frequency of vaginal temperature (°C) and respiration rate (breaths/min) and also describe the protocol of water access for the cows. Additionally, please include measurements of mean individual heat stress, if available. Are there any data regarding heat stress conditions that the cows were exposed to before the onset of the experiment? Please explain in greater detail the environmental conditions and the fluctuation of THI within 24h.

Au : Thank you for taking the tme to review our work. Procedures concerning respiration rates and vaginal temperatures measurments were described in depth in Fabris et al. [2020] and Dado-Senn et al. [2020]. As suggested more information on how we measured respiration rates and vaginal temperature is now provided (please see lines 104-109). Statistical analysis ran on vaginal temperature and respiration rates are presented at lines 242-243. Unfortunately, no environmental data was recorded before the start of the experiment. However, all cows were housed together, thus exposed to the same conditions. Exploring the daily evolution of THI is beyond the scope of the present study. However, THi remained aboved the heta stress threshold of 68 troughout the duration of the study.

Reviewer 4 Report

My assessment of the work under the title: “Endocrine signals altered by heat stress impact dairy cow mammary cellular processes at different stages of the dry period”

 I have read the paper carefully and come to the following remarks:

  • The work is well written, but there are a few comments to be taken into account.
  • In the abstract of line 29, the authors did not mention the composition of the basal liquid, although this is very necessary because many scientists only read the abstract. Therefore, the abstract must be clear and complete.
  • In the abstract, the oestrogen used had to be mentioned (17beta-estradiol).
  • The authors mentioned that the trial period was from May to November. The question is whether this trial period was the same trial period for both groups of cows that were under different temperature and humidity?
  • Unfortunately, not all of the words in Figure 5 can be read. I recommend enlarging the words so the reader can see them.
  • Physiologically it is known that under normal ambient temperature during the dry period (end of pregnancy) the hormone prolactin in the blood increases and progesterone decreases so that the lactogenesis starts. It has been better to discuss the work about it so that we can better understand the influence of heat stress on the mammary gland.
  • The reference list is good, but some references are not fully written. Some of them are written and in front of the last author name with the word "and" (as 18, 19, 23, 27, 29, 35, 38 and 45), the other references without and. Why this difference?
  • In addition, some references were written without end with page numbers such as 31, 34 and 38. Why?

I hope the authors can take my comments into account.

Best Regards

I wish you succsess

Author Response

R4:In the abstract of line 29, the authors did not mention the composition of the basal liquid, although this is very necessary because many scientists only read the abstract. Therefore, the abstract must be clear and complete.

Au : We understand your concern. However, the number of words is limited to 200 words by the journal. Therefore, we to limit the details of what we did to ensure compliance but still be understood by the readers. The readers can find the composition of the basal media in the materials and methods section.

R4:In the abstract, the oestrogen used had to be mentioned (17beta-estradiol).

Au:Changes we made according to the suggestion. Please line 30.

R4: The authors mentioned that the trial period was from May to November. The question is whether this trial period was the same trial period for both groups of cows that were under different temperature and humidity?

Au: Yes, it is the same trial period for both groups. Both treatments are subjected to the same environmental conditions. The only difference is that the cooled group have access to fans and soakers as described in the material and methods section.

R4: Unfortunately, not all of the words in Figure 5 can be read. I recommend enlarging the words so the reader can see them.

Au: Thank you for your comment. We double checked and all the words can be read in the edited version made by the journal.

R4: Physiologically it is known that under normal ambient temperature during the dry period (end of pregnancy) the hormone prolactin in the blood increases and progesterone decreases so that the lactogenesis starts. It has been better to discuss the work about it so that we can better understand the influence of heat stress on the mammary gland.

Au:Thank you for your comment, and agree that heat stress at parturition may influence the periparturient prolactin surge. However, the timing of the surge would be after the periods when our samples were collected, and thus beyond the scope of the present experiment.

R4:The reference list is good, but some references are not fully written. Some of them are written and in front of the last author name with the word "and" (as 18, 19, 23, 27, 29, 35, 38 and 45), the other references without and. Why this difference?

In addition, some references were written without end with page numbers such as 31, 34 and 38. Why?

Au: All references were formatted according to the journal’s guidelines.